# Effect of tempo on the age-related changes in temporal expectation driven by rhythms

**Zhihan Xu**[1]*, **Wenying Si**[1], **Yanna Ren**[2], **Yuqing Jiang**[1], **Ting Guo**[1]*

**1** Department of Foreign Language, Ningbo University of Technology, Ningbo, Zhejiang, China,
**2** Department of Psychology, Medical Humanities College, Guizhou University of Traditional Chinese Medicine, Guiyang, China

* guot1801@163.com (TG); xzhan22@163.com (ZX)

**Data Availability Statement:** All data have been publicly available on the [Open Science Framework] and can be accessed at [https://osf.io/yq2rz/].

## Abstract

Temporal expectation refers to the capacity to allocate resources at a particular point in time, enabling us to enhance our behavior performance. Empirical evidence indicates that, among younger adults, temporal expectation can be driven by rhythm (i.e., regular sequences of stimuli). However, whether there are age-related changes in rhythm-based temporal expectation has not been clearly established. Furthermore, whether tempo can influence the relationship between rhythm-based temporal expectation and aging remains unexplored. To address these questions, both younger and older participants took part in a rhythm-based temporal expectation task, engaging three distinct tempos: 600 ms (fast), 1800 ms (moderate), or 3000 ms (slow). The results demonstrated that temporal expectation effects (i.e., participants exhibited significantly faster responses during the regular trials compared to the irregular trials) were observed in both the younger and older participants under the moderate tempo condition. However, in the fast and slow tempo conditions, the temporal expectation effects were solely observed in the younger participants. These findings revealed that rhythm-based temporal expectations can be preserved during aging but within a specific tempo range. When the tempo falls within the range of either being too fast or too slow, it can manifest age-related declines in temporal expectations driven by rhythms.

## Introduction

Time processing plays a crucial role in multiple aspects of human cognition and behavior, including perceiving, measuring, and producing temporal intervals. It enables us to perceive and comprehend events in our environment accurately and aids in determining when to respond to different situations [1–3]. Numerous studies have reported age-related changes in time processing, with older adults demonstrating poorer performance in estimating intervals of a few seconds or hundreds of milliseconds [4–9]. These studies have mainly employed duration estimation tasks, where explicit and deliberate temporal processing is required, and participants overtly estimate the duration of specific time intervals [10–12].

However, timing is not limited to estimating the duration of ongoing events (duration estimation); it also involves anticipating when future events are likely to occur (temporal expectation). Fewer studies have focused on age-related differences in temporal expectation tasks, in

**Funding:** This research was funded by Zhejiang Provincial Philosophy and Social Sciences Planning Project (22NDQN280YB). URL:www.zjskw.gov.cn The funders had no role in study design, data collection and analysis, decision to publish, or preparation of the manuscript.

**Competing interests:** The authors have declared that no competing interests exist.

which timing is measured implicitly, without requiring conscious estimates of duration; participants tacitly anticipate the occurrence of future events at specific time points or intervals [13, 14]. Further research on age-related changes in temporal expectation tasks can provide valuable insights into the broader understanding of age-related alterations in time processing.

Temporal expectation signifies the skill to concentrate attention on a future critical specific time point, optimizing the processing of relevant stimuli that occur at the anticipated moment [14–16]. Empirical evidence shows that temporal expectation can be built up by various sources of temporal information, for instance, symbolic cues and regular rhythm, to accelerate motor preparation or improve perceptual discrimination [3, 17–19]. Temporal expectations induced by symbolic cues, which provide information on the duration of the cue-target interval, are believed to direct attention endogenously and voluntarily [20, 21]. In contrast, temporal expectation created by rhythms involving the isochronous presentation of a stimulus sequence is suggested to be generated automatically and exogenously [22, 23]. A wealth of research evidence supports the distinction between temporal expectations induced by symbolic cues and those created by rhythms [24–28]. However, studies on temporal expectation have primarily concentrated on younger adults, and the extent to which temporal expectation is preserved in healthy aging remains elusive.

Zanto and colleagues (2011) manipulated temporal expectations in younger and older adults using visual symbolic cues. The letter 'S' or 'L' indicated to participants that a target would appear after either 600 ms or 1400 ms. This manipulation was applied across several tasks. They revealed that older adults were unable to successfully utilize symbolic temporal cues to allocate attention at the right instant. Their results demonstrated that younger adults exhibited faster reaction times when the target appeared after a validly cued preparatory interval compared to neutral cues. This trend persisted across tasks, ranging from simple detection tasks to more complex go/no-go discrimination and forced-choice discrimination tasks. In contrast, older participants failed to benefit from a predictive temporal cue across all the above task conditions. They proposed a deficit for older people in temporal expectations driven by symbolic cues [29]. In contrast, Chauvin and colleagues (2016) demonstrated that both young and older participants could gain an advantage from auditory temporal cues to accelerate response time and improve accuracy for the temporally predictable target. They proposed that symbolic cue-based orienting attention to appropriate moments could be preserved in normal aging [30]. Moreover, as mentioned above, currently, the limited research on the aging effects of temporal expectations has predominantly focused on those formed by symbolic cues. However, there is still a notable dearth of studies exploring the aging effects of temporal expectations induced by rhythm.

Temporal expectation driven by rhythm involves more automatic processing and demands fewer cognitive resources, such as attention and working memory, compared to temporal expectation triggered by symbolic cues. Despite the findings by Zanto and colleagues (2011) suggesting a deficit in temporal expectation induced by symbolic cues in older adults, it is widely acknowledged that as age increases, there is a decrease in both working memory abilities and the control of attention [7, 31–34]. Drawing from these findings, it has been doubted whether the deficits in temporal expectation driven by symbolic cues can be genuinely attributed to a decline in temporal information processing in older adults or should be rather considered as a side effect of the reduced cognitive functioning in this population. Therefore, opting for rhythm-based temporal expectation tasks that utilize little controlled attentional resources, or at a minimum, entail an exceedingly low cognitive load, for further research would provide a more accurate reflection of older adults' ability to predictive timing.

In addition, previous studies on temporal expectations created by rhythms have predominantly concentrated on tempos of less than 1 second (sub-second scale) [17, 19, 22, 35–38].

Tempo refers to the rate or pace of a stimulus sequence, indicating how fast or slow it is, typically stated as the time interval between successive stimuli. Several studies have proposed different mechanisms for measuring durations in the sub-second and supra-second ranges. It has been proposed that time measurements in the sub-second scale are automatic, whereas those in the supra-second scale demand the engagement of cognitive control [39–42]. Furthermore, in our previous study, we investigated the impact of tempo on the mechanism underlying temporal expectation triggered by rhythms. Participants performed a rhythm-based temporal expectation task alongside a working memory task. The findings demonstrated that temporal expectation effects remained intact in the presence of dual-task interference under the fast-tempo condition while being compromised under the slow-tempo condition. This suggests that fast tempo-induced rhythmic temporal expectation relies more on automatic processing, while slower tempo-driven temporal expectation may involve more controlled processing [43]. Therefore, a significant gap in the existing research is exploring whether the relation between rhythm-based temporal expectation and aging can be influenced by tempo.

In summary, we have two primary goals in our present study. Our foremost objective is to assess whether older adults have a deficit in rhythm-based temporal expectations. A secondary aim of the study is to evaluate whether the effects of aging on rhythm-based temporal expectation may vary with different tempos. For this purpose, we included three tempos: 600 ms for the fast tempo, 1800 ms for the moderate tempo, and 3000 ms for the slow tempo. These tempo variations allow us to investigate the relationship between tempo and rhythm-based temporal expectation in the aging population.

## Methods

### Participants

An a priori power analysis was performed using G*Power 3.1.9.2 [44] for a repeated-measures analysis of variance (ANOVA), within-between interaction. We adopted an effect size of 0.2, referencing the findings from Chauvin et al., 2016 [30], while setting alpha at 0.05 and power at 0.8. According to these parameters, each age group required a minimum of 14 participants. Consequently, our study recruited 18 young adults (9 males and 9 females, age range 18–23 years old, mean: 19.9) and 18 older adults (9 males and 9 females, age range 60–74 years old, mean: 65), ensuring substantial statistical power for achieving the primary goal of our research.

The young adult participants were recruited from Ningbo University of Technology, while the older adult participants were recruited from the social circles of the students' acquaintances. All participants self-reported as right-handed and had normal or corrected vision. They had no history of psychiatric or neurological disorders and were not taking any vasoactive or psychotropic medications. In the three years preceding the experiment, all participants had no previous experience or formal training in professional music and did not actively play any musical instruments. Before enrollment, all participants provided written informed consent for the study, which received approval from the institutional ethics committee. The recruitment period for participants was from October 20, 2022, to December 16, 2022. Monetary compensation was provided to the participants for their participation. To exclude participants suffering from dementia, the MMSE [45] was administered to all older participants. Notably, all the older participants scored above the cut-off threshold of 27 on the MMSE (mean score = 29.06, SD = 1).

### Apparatus and stimuli

The E-prime software [46] served as the platform for presenting stimuli and collecting reaction time (RT) from participants. Each trial consisted of a fixation point, stimulus sequence, and

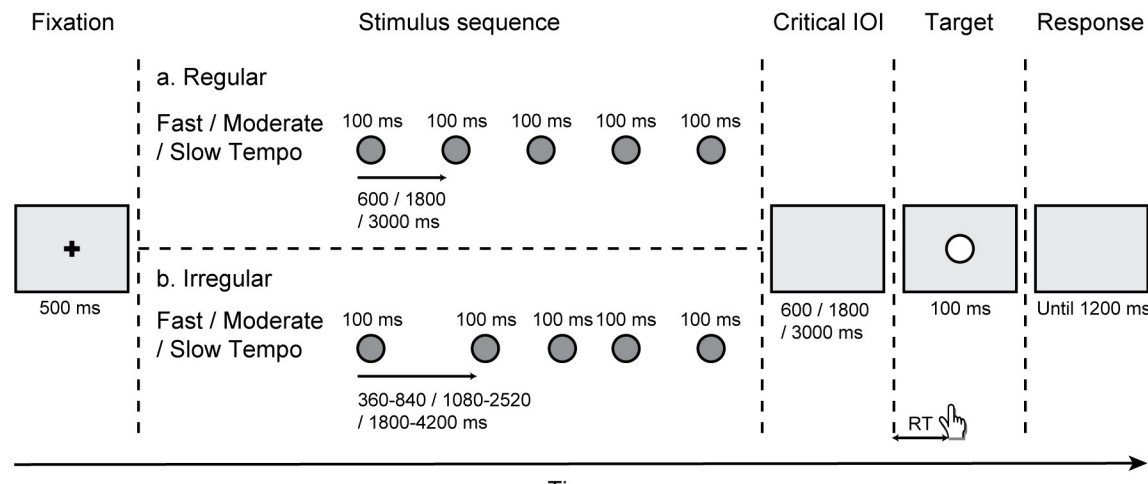

**Fig 1. Schematic representation of events in a trial.** Participants were required to respond to a white circle target that appeared following either a regular (a) or an irregular (b) rhythm, with tempo options of fast (600 ms), moderate (1800 ms) or slow (3000 ms).

target. All stimuli were displayed at the center of a 27-inch monitor against a grey background (RGB = 180, 180, 180) with a resolution of 1980*1080 pixels. The viewing distance was 60 cm from the screen's center.

The fixation point was represented by a black "+" symbol (1.2˚ × 1.2˚ of visual angle) displayed for 500 ms. The stimulus sequence comprised five consecutive circles, with each circle presented for 100 ms. These circles were solid grey (diameter = 2˚; RGB = 100, 100, 100). The sequence could have either a regular or irregular timing pattern. In the regular sequence, the grey circles were presented with inter-onset intervals (IOIs) of 600 ms (fast tempo), 1800 ms (moderate tempo), or 3000 ms (slow tempo), randomization between blocks. For the irregular sequence, the IOIs could be 360, 480, 600, 720, or 840 ms (fast tempo); 1080, 1440, 1800, 2160, or 2520 ms (moderate tempo); or 1800, 2400, 3000, 3600, or 4200 ms (slow tempo). The order of these five intervals was arranged randomly during the trials. The stimulus intended as the target was a white circle (diameter = 2˚) displayed for 100 ms (see Fig 1).

## Procedure and task

Participants were comfortably seated on a chair in a quiet and softly lit room. To ensure stability, their head positions were secured using a chin rest. Clear instructions were given to participants through both written and verbal means. Each participant underwent a practice block followed by 15 experimental blocks. These experimental blocks consisted of three tempo categories: 5 fast tempo blocks, 5 moderate tempo blocks, and 5 slow tempo blocks. The presentation order of these experimental blocks was randomly intermixed. Additionally, each block contained 40 trials, evenly divided into 20 regular trials and 20 irregular trials, all of which were randomly delivered. Fig 1 provides a schematic of the task. Each trial commenced with the presentation of a fixation point for 500 ms. Subsequently, a randomly selected sequence, either regular or irregular, was displayed with equal probabilities. A white circle target emerged following the sequence, with a critical IOI of 600 ms (fast tempo), 1800 ms (moderate tempo), or 3000 ms (slow tempo). Participants were directed to press the left button of mouse using their dominant index finger as quickly as possible upon the presentation of the target. The target appeared for 100 ms, then was replaced by a blank grey background. This grey

background remained until the participant responded, or for a maximum of 1200 ms, to ensure a timely response.

In a fifth of the trials (20%), no target stimulus was displayed, and these were referred to as catch trials, where no response was required. Catch trials were introduced to prevent the influence of a "hazard function," in which expectations were formed relying on the conditional probability that the stimulus would occur given that it had not yet occurred [47].

### Design and data analysis

The experiment consisted of a $2_{group}$ (Young, Old) × $3_{tempo}$ (Fast, Moderate, Slow) × $2_{rhythm}$ (Regular, Irregular) mixed factor design. The group variable was considered a between-participants factor, while the tempo and rhythm variables were within-participants factors. Our primary outcome variable was the mean RT for correct responses in each condition. The RT signifies the duration from the initiation of the target presentation to the motor response. For each participant, RTs exceeding three standard deviations from the mean were excluded from the analysis, calculated separately for each dependent variable (tempo and rhythm) [30]. The proportion of excluded trials due to outliers was found to be similar between the younger group (1.36% and the older group (1.52%). According to the results of the two-sample t-tests: t (34) = 0.456, $p$ = 0.488, no statistically significant difference was observed between the two groups. To account for potential violations of sphericity, Greenhouse-Geisser corrections were applied, adjusting the degrees of freedom accordingly. The statistical significance level was set at $p < .05$, and the effect size ($\eta_p^2$) estimates are also provided. All statistical analyses were carried out utilizing the SPSS version 19.0 software (SPSS, Tokyo, Japan), ensuring consistency throughout the study.

### Results

Table 1 provides the detailed mean RTs for each condition. The 3-way mixed-design ANOVA with factors of 2 group (Young, Old) × 3 tempo (Fast, Moderate, Slow) × 2 rhythm (Regular, Irregular) demonstrated a significant main effect of the group [F (1,34) = 19.058, $p < 0.001$, $\eta_p^2 = 0.359$]. Younger adults exhibited faster RTs (281.84 ms) than older adults (353.25 ms). A significant main effect of tempo was observed [F (2,68) = 80.708, $p < 0.001$, $\eta_p^2 = 0.704$], indicating faster RTs in the fast tempo condition (285.34 ms) compared to the moderate tempo condition (323.17 ms) and slow tempo condition (344.12 ms), with all $p < 0.001$. No significant interaction effect was found between tempo and group [F (2,68) = 1.135, $p = 0.315$], suggesting that both the younger and older groups exhibited a similar trend. Specifically, as the tempo slowed down, RTs progressively lengthened: fast< moderate < slow, with all $p \leq .002$ (see Fig 2).

The main effect of rhythm was found to be significant [F (1,34) = 41.656; $p < 0.001$; $\eta_p^2 = 0.551$], indicating that RTs were faster for regular trials compared to irregular trials. The interaction between rhythm and group was also significant [F (1,34) = 4.401, $p = 0.043$, $\eta_p^2 =$

**Table 1. Mean RTs (ms) for each group (Young, Old), tempo (Fast, Moderate, Slow) and rhythm condition (regular and irregular).**

|  | Younger group | | | Older group | | |
|---|---|---|---|---|---|---|
|  | **Fast** | **Moderate** | **Slow** | **Fast** | **Moderate** | **Slow** |
| Regular | 239(11) | 281(13) | 299(11) | 315(11) | 353(15) | 379(14) |
| Irregular | 267(11) | 295(13) | 310(11) | 321(9) | 364(15) | 388(13) |

Values in parentheses are standard errors of the mean.

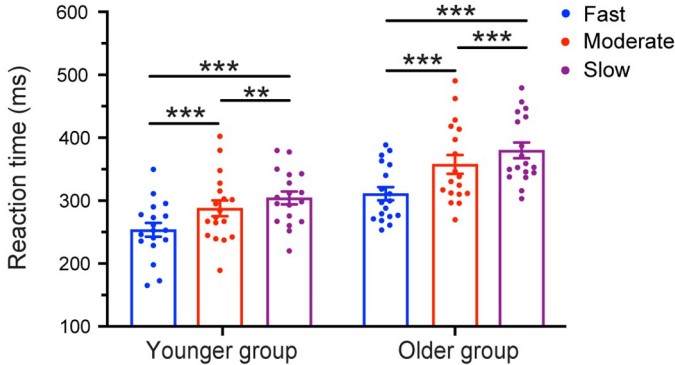

**Fig 2. Mean RTs for the younger and older groups in each tempo condition (Fast, Moderate, Slow), collapsed across rhythm condition.** The error bars represent the standard error of the mean. Statistical significance is denoted by \*\*\**p*<0.001 and \*\**p* < 0.01.

0.115]. Most notably, a significant three-way interaction between tempo, rhythm, and group was observed [$F_{(2,68)} = 3.406$, $p = 0.039$, $\eta_p^2 = 0.091$]. Follow-up pairwise comparisons (corrected using Bonferroni correction) revealed that in the moderate tempo condition, mean RTs for regular trials were significantly faster compared to the irregular trials in both the younger group ($p = 0.001$) and the older group ($p = 0.006$). However, under the fast and slow tempo conditions, the mean RTs for regular trials were significantly faster compared to irregular trials only in the younger group (fast: $p < 0.001$; slow: $p = 0.028$), while no significant difference was observed between regular and irregular trials in the older group (fast: $p = 0.223$; slow: $p = 0.052$) (see Fig 3).

## Discussion

The present study aimed to investigate the effectiveness of regular rhythm in driving temporal expectations in older adults. Additionally, the study included three tempo conditions: 600 ms

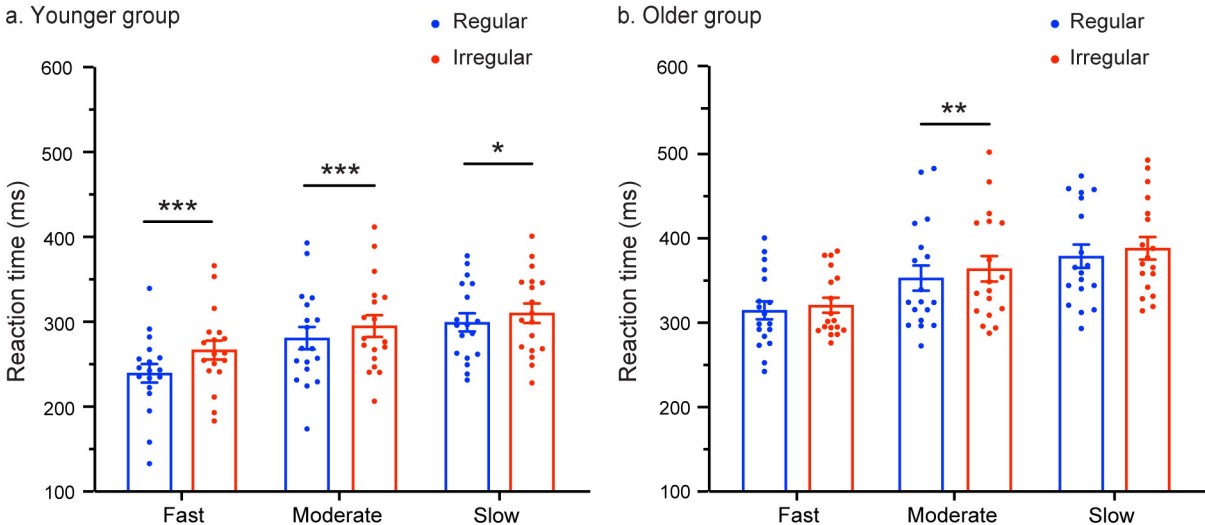

**Fig 3. Mean RT as a function of the tempo condition (fast, moderate, and slow) and rhythm (regular and irregular) for the younger (a) and older (b) groups.** The error bars represent the standard error of the mean. Statistical significance is denoted by \*\*\**p* < 0.001, \*\* *p* < 0.01, \* *p* < 0.05.

for the fast tempo, 1800 ms for the moderate tempo, and 3000 ms for the slow tempo, in order to explore whether the relation between rhythm-based temporal expectation and aging can be influenced by tempo. The results revealed that under the moderate tempo condition, both the younger and older groups showed temporal expectation effects, with mean RTs for regular trials significantly faster than for the irregular trials. However, during the fast and slow tempo conditions, the temporal expectation effects were only observed in the younger group.

Both age groups demonstrated a consistent finding that RTs increased gradually with the decrease in tempo. This observation can be partly interpreted as the shorter interval enabling more accurate and confident anticipation of the target's onset moment. Conversely, a longer interval leads to predictions that are more variable and less precise, accompanied by increased uncertainty, which in turn results in a slower RT [48]. Additionally, these results align with the outcomes of our previous research, where younger adults exhibited faster RTs in response to fast-paced rhythm compared to slow-paced rhythm [27, 28, 43]. The possibility should also be considered that the mechanism of rhythm-based temporal expectations varies with tempo. In our previous study (2021), we proposed that rhythm-based temporal expectation created by fast tempo relied on automatic processing. However, temporal expectations created by a slower tempo might engage more controlled processing, and memory-based strategies are employed to supplement the formation of temporal expectations [43]. Furthermore, recent studies have suggested that automatically processed rhythm-based temporal expectations are more effective in optimizing behavioural performance compared to memory-based temporal expectations that require more controlled attentional resources [27, 28, 49]. The results of the present study corroborate previous reports that RTs lengthen with slowing down of tempo. These findings extend the previous research by revealing that tempo has a similar impact on rhythm-based temporal expectations in older adults as it does in younger adults.

In the context of fast tempo conditions, solely the younger adults demonstrated significantly faster RTs in regular trials when contrasted with irregular trials; no significant difference was observed among the older adults. Revealing that, under fast tempo conditions, older participants have a deficit in rhythm-based temporal expectation. Our results are consistent with previous studies, where an age-related difference is observed in producing or continuing a rhythm [50]. Turgeon and Wing (2012) conducted a study involving healthy adults aged 19 to 98, administering a set of unpaced timing tasks (also reported in McAuley et al., 2006), such as spontaneous motor tapping (SMT) task, where participants were instructed to tap at the pace they feel most comfortable, and synchronize-continue (SC) task, where participants initially tapped in sync with an external stimulus and were then instructed to sustain the same tapping pace after the stimulus was no longer present. Their results showed that as age increased, tapping rates became longer and more variable, providing evidence for an age-related decline in temporal processing [50, 51].

However, as proposed by JT Coull (2008) [24], tasks such as the synchronize-continue (SC) and similar unpaced timing tasks require subjects to represent inter-stimulus intervals (ISIs) through sustained, delayed or periodic motor actions. Moreover, these tasks necessitate subjects to offer a precise estimate of elapsed time, placing them within the realm of explicit timing. Conversely, rhythm-based temporal expectation engages participants even when they are unaware they are processing time. This occurs whenever sensorimotor information is temporally structured and can be utilized to anticipate the duration of upcoming events categorizing it as implicit timing. Dissociations between explicit and implicit timing have been demonstrated at both neurological and behavioural levels [24, 52–54]. Furthermore, it has been suggested that explicit timing might recruit more attentional control than implicit timing [55]. Hence, the outcomes of the current rhythm-based temporal expectation tasks, seemingly

minimizing the engagement of memory and attention processes, appear to indicate a 'true' age-related distinction in temporal processing.

In addition, it's worth noting that previous research has predominantly concentrated on tempos shorter than 1 second, overlooking the potential influence of tempo on age-related changes in rhythm-based temporal expectation. In contrast, our current study delves into a broader spectrum, encompassing three distinct tempos that span from sub-second to supra-second intervals. Intriguingly, our results indicated a dynamic relationship between aging and rhythm-based temporal expectations, one that varies with alterations in tempo, unveiling meaningful and thought-provoking insights in this regard.

Notably, the age-related decrease in rhythm-based temporal expectation under the fast tempo condition vanished in the moderate tempo condition. RTs for regular trials were significantly faster than for irregular trials in both the younger and older groups. These findings could potentially find an explanation through the compensation-related utilization of neural circuits hypothesis (CRUNCH) put forth by Reuter-Lorenz and colleagues [56–59]. According to this hypothesis, older adults activate supplementary brain regions and networks compared to younger adults when undertaking identical cognitive tasks. This added engagement aims to enable older adults to perform at a level similar to, or nearly on par with the effectiveness of young adults, in order to compensate for the age-related diminishes in brain structures and cognitive functions. As revealed by preceding studies, the rhythm-based temporal expectation triggered by a fast tempo was found to rely on automatic processing. Nevertheless, as the tempo gradually decelerates, it necessitates the involvement of controlled mechanisms [43]. Consequently, age-related differences observed in the rhythm-based temporal expectation during fast tempo, where cognitive involvement is minimized, provide a minimal chance for alternative networks /processes to intervene. Conversely, as the tempo decelerates and enters the 'cognitive realm,' older adults may employ compensatory mechanisms to maintain their performance [60]. As a result, no aging-related differences were found under moderate tempo conditions.

Even more captivatingly, age-related changes apparent under the fast tempo condition vanish under the moderate tempo condition, but reappear in the slow condition as age-related differences. As depicted in Fig 3, it can be observed that among young adults, the temporal expectation effect, characterized by markedly faster mean RTs for regular trials compared to irregular ones, holds true across all three tempo conditions. In contrast, among older adults, this temporal expectation effect emerged solely within the moderate tempo condition. We speculate that the reason for this phenomenon might be, as suggested by Paulsen and Collages (2004), that when the load of either cognitive demands or disease-related physical declines surpasses their compensation threshold, those alternative processes or networks would experience a "break down", resulting in the resurgence of age-related declines [61].

Additionally, it's important to emphasize that the difference between regular and irregular trials in the older group under the slow tempo condition is marginally significant ($p = 0.052$). We hypothesize that the exclusion of individuals in the very old age bracket from the older groups could potentially account for this, especially considering that some functions tend to exhibit age-related differences only in the very late stages of life [62]. To explore this aspect more comprehensively, forthcoming studies could contemplate validating these observations by enrolling participants spanning a diverse range of ages.

## Conclusion

In conclusion, our present study demonstrated that both the younger and older groups exhibited temporal expectation effects under the moderate tempo condition. However, in the fast

and slow tempos, only the younger group manifested the temporal expectation effect, with no corresponding manifestation in the older group. These results indicated that rhythm-induced temporal expectations can be preserved in aging only within a specific tempo range. When the tempo falls within the range of either being too fast or too slow, it can manifest age-related declines in rhythm-based temporal expectations. These outcomes revealed that the relationship between aging and rhythm-based temporal expectations may not be immutable; instead, it undergoes modulation influenced by tempo variations.

## Author Contributions

**Conceptualization:** Zhihan Xu, Ting Guo.

**Data curation:** Zhihan Xu, Wenying Si.

**Formal analysis:** Zhihan Xu, Ting Guo.

**Funding acquisition:** Zhihan Xu.

**Investigation:** Zhihan Xu, Wenying Si.

**Methodology:** Zhihan Xu, Yanna Ren, Ting Guo.

**Project administration:** Zhihan Xu, Ting Guo.

**Resources:** Zhihan Xu.

**Supervision:** Zhihan Xu, Ting Guo.

**Validation:** Zhihan Xu, Yanna Ren.

**Visualization:** Zhihan Xu.

**Writing – original draft:** Zhihan Xu.

**Writing – review & editing:** Zhihan Xu, Yuqing Jiang, Ting Guo.

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
