## [Decision Letter · Decision Letter 0]

13 Nov 2023

PONE-D-23-26702Effect of tempo on the age-related changes in temporal expectation driven by rhythmsPLOS ONE

Dear Dr. Xu,

Thank you for submitting your manuscript to PLOS ONE. After careful consideration, we feel that it has merit but does not fully meet PLOS ONE’s publication criteria as it currently stands. Therefore, we invite you to submit a revised version of the manuscript that addresses the points raised during the review process.

Please, when preparing your revised manuscript. take into account the comments and concerns of the reviewer, in particular those related to conceptual and methodological improvements in your presentation that will contribute to enhance the scope and readability of your work,

We look forward to receiving your revised manuscript.

Kind regards,

Enrique Hernandez-Lemus, Ph.D.

Academic Editor

PLOS ONE

Journal Requirements:

Reviewers' comments:

Reviewer's Responses to Questions

**Comments to the Author**

1. Is the manuscript technically sound, and do the data support the conclusions?

Reviewer #1: Partly

2. Has the statistical analysis been performed appropriately and rigorously? 

Reviewer #1: No

3. Have the authors made all data underlying the findings in their manuscript fully available?

Reviewer #1: No

4. Is the manuscript presented in an intelligible fashion and written in standard English?

Reviewer #1: Yes

5. Review Comments to the Author

Reviewer #1: The study found that rhythm-based temporal expectations can be preserved during aging but within a specific tempo range. When the tempo falls within the range of either being too fast or too slow, it can manifest age-related declines in temporal expectations driven by rhythms. The authors argue that temporal expectation effects were observed in both the younger and older participants under the moderate tempo condition. However, the temporal expectation effects were solely observed in the younger participants in the fast and slow tempo conditions. However, the following commentaries address some concerns.

Main concerns,

Some methodologies related to reaction vs. motor times, and the means for measuring them, are not clear.

Bar plots are not the most optimal way of showing data for which the authors’ findings deserve better visual understanding.

These and other concerns are thoroughly given below.

Line 66 “A wealth of research evidence supports the distinction between these two types of temporal expectations [24-28]”.

It seems that the authors specifically refer to the difference between temporal duration and time intervals. A paper from Duarte F, 2017 is closely related to the ms. Additionally, some fundamental references from Profs. Merchant, Soto-Faraco, and Bounomano’s groups about timing and temporal estimation appear to be missing here.

L 72 “visual symbolic cues”

please explain what are symbolic cues

L.130-133 What was the statistical test for deciding the correct number of participants in each group?

L.175 “block containing 40 trials”.

Did each participant perform all 15 blocks? What did each block consist of? Perhaps more important, where blocks presented intermingled?

L.178-179 times are not consistent with those of fig1

L.180 button as quickly as possible upon the presentation of the target. What were the characteristics of the button, where was it situated, and was there a second right button? And what hardware was used for the task’s control and readout.

L.181 “In a fifth of the trials (20%)”

Why 20% of trials? This is not a counterbalanced number of trials.

L.182 “to as catch trials”.

Catch trials refer to trials of responses not influenced by the tests. However, it is required to get a response. How these responses were created if no target stimulus was presented?

L.185 “response, participants were given a maximum of 1200 ms to initiate their response”.

For the response window, was the 100 ms target stimulus present? Again, where were responses delivered?

L.192 “target presentation to the execution of the motor response”

That would be the motor times. Reaction times do not entirely depend on the speed of motor responses but on the reaction to a perceived variable. In this regard, the authors do not clarify what the initial position of the response hand was (assuming responses were delivered with a hand). Here, reaction times should have been the times from the appearance of the target to the movement initiation. Moreover, since there is a group of elders, some other response metrics should have been considered, e.g., eye movements, and changes in pupil diameter.

Results

Given the small size of groups, it would be optimal to observe the participants' performance in a box plot and not the overall population result in a bar plot.

Line 214 (see Fig 2). It is hard to believe that variance did not increase with tempo conditions. Again, can the authors please present box plots where individuals' performance can be observed?

Line 223. The effect of each of the conditions within each group seems a consequence of presenting blocks of trials of each condition separately. Again, were conditions delivered in blocks of consecutive trials?

6. PLOS authors have the option to publish the peer review history of their article (what does this mean?). If published, this will include your full peer review and any attached files.

Reviewer #1: No

---

## [Author Response · Author response to Decision Letter 0]

4 Dec 2023

Manuscript Number: PONE-D-23-26702

Title: Effect of tempo on the age-related changes in temporal expectation driven by rhythms

Response to Reviewer 1

We express our sincere gratitude for your valuable and detailed suggestions. Following your insightful feedback, we have carefully revised the original manuscript. Below, we have listed all the sections that were modified, marking these changes in the manuscript with a grey background for easy identification. Due to the major revisions, we hope our manuscript has been improved greatly. Once again, we deeply appreciate your comments and guidance.

1) Line 66 “A wealth of research evidence supports the distinction between these two types of temporal expectations [24-28]”. It seems that the authors specifically refer to the difference between temporal duration and time intervals. A paper from Duarte F, 2017 is closely related to the ms. Additionally, some fundamental references from Profs. Merchant, Soto-Faraco, and Bounomano’s groups about timing and temporal estimation appear to be missing here.

Response:

Thank you for your insightful feedback. We apologize for any confusion caused by our initial description and appreciate your assistance in clarifying this matter.

In our original text, "the distinction between these two types of temporal expectations," we were actually referring to "the difference between temporal expectations induced by symbolic cues and those created by rhythms," not "the difference between temporal duration and time intervals."

Indeed, as per the references you provided about timing and temporal estimation, there has been proposed a distinction between temporal estimation and temporal expectation. Temporal estimation tasks explicitly measure timing, requiring the overt measurement and registration of elapsed time in a motor act or sensory event. In contrast, temporal expectation tasks measure timing implicitly, without necessitating conscious duration estimates; participants intuitively anticipate the occurrence of future events at specific time points or intervals.

However, what we intend to express in this section is not the difference between temporal estimation and temporal expectation. Rather, it is the difference between temporal expectations induced by symbolic cues and those created by rhythms.

We apologize again for any previous lack of clarity and are grateful for your input in improving our manuscript. The following revisions have been made to enhance clarity and precision in our description. We hope these adjustments effectively address the issues raised.

L.66 – 68

A wealth of research evidence supports the distinction between temporal expectations induced by symbolic cues and those created by rhythms.

2) L 72 “visual symbolic cues”. please explain what are symbolic cues

Response:

Thank you for pointing out the areas where our description was lacking. We have made the following modifications based on your suggestions:

L.71 – 74

Zanto and colleagues (2011) manipulated temporal expectations in younger and older adults using visual symbolic cues. The letter 'S' or 'L' indicated to participants that a target would appear after either 600 ms or 1400 ms. This manipulation was applied across several tasks.

3) L.130-133 What was the statistical test for deciding the correct number of participants in each group?

Response:

We apologize for the oversight in our initial description. In fact, we determined the number of participants in each group based on previous related studies on temporal expectation, as well as calculations made using G-power. 

Firstly, we briefly listed the number of participants in several studies cited in our manuscript. 

[1] 18 younger participants and 18 older participants took part in the experiment (Chauvin, Gillebert, Rohenkohl, Humphreys, & Nobre, 2016).

[2] Sixteen participants gave informed consent to take part in the experiment (A. Correa & Nobre, 2008). 

[3] Fifteen participants took part in Experiment 1, and fourteen in Experiment 2 (Breska & Deouell, 2014). 

[4] Sixteen undergraduates participated in the study (Sanabria & Correa, 2013). 

[5] Eighteen right-handed participants took part voluntarily in this experiment (Rohenkohl, Coull, & Nobre, 2011).

[6] In this study, 34 healthy, right-handed students fromOkayama University took part as volunteers. Participants were randomly assigned to the single-task (17 participants) and dual-task (17 participants) conditions (Xu, Ren, Misaki, Wu, & Lu, 2021). 

Additionally, thank you for highlighting this oversight. We have revised the original manuscript to include a description of our use of G-power.

L.132 – 139

An a priori power analysis was performed using G*Power 3.1.9.2[44] for a repeated-measures analysis of variance (ANOVA), within-between interaction. We adopted an effect size of 0.2, referencing the findings from Chauvin et al., 2016 [45], while setting alpha at 0.05 and power at 0.8. According to these parameters, each age group required a minimum of 14 participants. Consequently, our study recruited 18 young adults (9 males and 9 females, age range 18–23 years old, mean: 19.9) and 18 older adults (9 males and 9 females, age range 60–74 years old, mean: 65), ensuring substantial statistical power for achieving the primary goal of our research.

· Breska, A., & Deouell, L. Y. (2014). Automatic bias of temporal expectations following temporally regular input independently of high-level temporal expectation. Journal of cognitive neuroscience, 26(7), 1555-1571. 

· Chauvin, J. J., Gillebert, C. R., Rohenkohl, G., Humphreys, G. W., & Nobre, A. C. (2016). Temporal orienting of attention can be preserved in normal aging. Psychology and aging, 31(5), 442. 

· Correa, A., & Nobre, A. C. (2008). Neural modulation by regularity and passage of time. Journal of Neurophysiology, 100(3), 1649-1655. 

· Rohenkohl, G., Coull, J. T., & Nobre, A. C. (2011). Behavioural dissociation between exogenous and endogenous temporal orienting of attention. PloS one, 6(1), e14620. 

· Sanabria, D., & Correa, Á. (2013). Electrophysiological evidence of temporal preparation driven by rhythms in audition. Biological psychology, 92(2), 98-105. 

· Xu, Z., Ren, Y., Misaki, Y., Wu, Q., & Lu, S. (2021). Effect of Tempo on Temporal Expectation Driven by Rhythms in Dual-Task Performance. Frontiers in Psychology, 12, 755490.

4) L.175 “block containing 40 trials”. Did each participant perform all 15 blocks? What did each block consist of? Perhaps more important, where blocks presented intermingled?

Response:

We sincerely appreciate you for identifying the key issues in our description. We apologize for any lack of clarity in our original manuscript. Following your suggestions, we have revised our manuscript as follows, hoping that these changes have effectively addressed the concerns you highlighted.

L.180 – 185

Each participant underwent a practice block followed by 15 experimental blocks. These experimental blocks consisted of three tempo categories: 5 fast tempo blocks, 5 moderate tempo blocks, and 5 slow tempo blocks. The presentation order of these experimental blocks was randomly intermixed. Additionally, each block contained 40 trials, evenly divided into 20 regular trials and 20 irregular trials, all of which were randomly delivered. 

5) L.178-179 times are not consistent with those of fig1

Response:

Thank you for pointing out this issue. Based on your recommendation, we have revised the manuscript as follows:

L.188 – 190

A white circle target emerged following the sequence, with a critical IOI of 600 ms (fast tempo), 1800 ms (moderate tempo), or 3000 ms (slow tempo).

6) L.180 button as quickly as possible upon the presentation of the target. What were the characteristics of the button, where was it situated, and was there a second right button? And what hardware was used for the task’s control and readout.

Response:

We apologize for not fully expressing this in our description and thank you for pointing out this issue. In fact, the participants were instructed to keep their dominant index finger on the left mouse button throughout the experiment, and were instructed to press the left mouse button using their dominant index finger as quickly as possible upon the presentation of the target. This button-pressing method is commonly used in previous related studies, for example:

[1] “Participants had to respond to the onset of the cross by pressing the left mouse button with their right hand as quickly as possible”(Capizzi, Martín-Signes, Coull, Chica, & Charras, 2023).

[2] “The participant was asked to use this information to respond to detection of the target as quickly as possible, by pressing the left side of an external mouse” (Johnson, Burrowes, & Coull, 2015).

[3] “The participants were instructed to respond as quickly as possible after the presentation of the target by pressing the left button with their index fingers while avoiding anticipation”(Xu et al., 2021).

[4] “Subjects had to detect any of the two letters—which appeared with identical probability (P = 0.5)—by pressing the right button of the mouse with their dominant hand”(Monica Trivino, Correa, Arnedo, & Lupianez, 2010).

Thank you for pointing out the deficiencies in our description. We have revised the original manuscript as follows:

L.190 – 192

Participants were directed to press the left button of mouse using their dominant index finger as quickly as possible upon the presentation of the target.

· Capizzi, M., Martín-Signes, M., Coull, J. T., Chica, A. B., & Charras, P. (2023). A transcranial magnetic stimulation study on the role of the left intraparietal sulcus in temporal orienting of attention. Neuropsychologia, 184, 108561. 

· Johnson, K. A., Burrowes, E., & Coull, J. T. (2015). Children can implicitly, but not voluntarily, direct attention in time. PloS one, 10(4), e0123625. 

· Trivino, M., Correa, A., Arnedo, M., & Lupianez, J. (2010). Temporal orienting deficit after prefrontal damage. Brain, 133(4), 1173-1185. 

· Xu, Z., Ren, Y., Guo, T., Wang, A., Nakao, T., Ejima, Y., . . . Wu, Q. (2021). Temporal expectation driven by rhythmic cues compared to that driven by symbolic cues provides a more precise attentional focus in time. Attention, Perception, & Psychophysics, 83, 308-314. 

7) L.181 “In a fifth of the trials (20%)” Why 20% of trials? This is not a counterbalanced number of trials.

Response:

First, catch trials were introduced to prevent the influence of a “hazard function,” in which expectations were formed relying on the conditional probability that the stimulus would occur given that it had not yet occurred.

Correa et al. (2006) implemented catch trials in their experiments, utilizing a range of proportions: 0%, 12.5%, 25%, and 50% of trials. The results showed that catch trials at proportions of 12.5%, 25%, and 50% of the total trials effectively prevented the influence of the "hazard function."

Therefore, the methodology of including catch trials has been widely adopted, typically with proportions exceeding 12.5%, as evidenced in studies such as 12.5% (Breska & Deouell, 2014, 2017), 16.7% (Sanabria & Correa, 2013), and 20% (Correa & Nobre, 2008; Trivino, Correa, Arnedo, & Lupianez, 2010; Xu, Ren, Misaki, Wu, & Lu, 2021). In our current study, we chose a relatively stable proportion of 20% for catch trials, which effectively helped us prevent the influence of a “hazard function”.

· Breska, A., & Deouell, L. Y. (2014). Automatic bias of temporal expectations following temporally regular input independently of high-level temporal expectation. Journal of cognitive neuroscience, 26(7), 1555-1571. 

· Breska, A., & Deouell, L. Y. (2017). Neural mechanisms of rhythm-based temporal prediction: Delta phase-locking reflects temporal predictability but not rhythmic entrainment. PLoS biology, 15(2), e2001665. 

· Correa, A., Lupiáñez, J., & Tudela, P. (2006). The attentional mechanism of temporal orienting: Determinants and attributes. Experimental Brain Research, 169, 58-68. 

· Correa, A., & Nobre, A. C. (2008). Neural modulation by regularity and passage of time. Journal of Neurophysiology, 100(3), 1649-1655. 

· Sanabria, D., & Correa, Á. (2013). Electrophysiological evidence of temporal preparation driven by rhythms in audition. Biological psychology, 92(2), 98-105. 

· Trivino, M., Correa, A., Arnedo, M., & Lupianez, J. (2010). Temporal orienting deficit after prefrontal damage. Brain, 133(4), 1173-1185. 

· Xu, Z., Ren, Y., Misaki, Y., Wu, Q., & Lu, S. (2021). Effect of Tempo on Temporal Expectation Driven by Rhythms in Dual-Task Performance. Frontiers in Psychology, 12, 755490. 

8) L.182 “to as catch trials”. Catch trials refer to trials of responses not influenced by the tests. However, it is required to get a response. How these responses were created if no target stimulus was presented?

Response:

We sincerely apologize for our oversight in describing the catch trials. Indeed, in catch trials, since the target was not presented, participants were not required to respond.

Thank you for pointing out this issue. We have made the following amendments to the original manuscript:

L.195 – 196

In a fifth of the trials (20%), no target stimulus was displayed, and these were referred to as catch trials, where no response was required.

9) L.185 “response, participants were given a maximum of 1200 ms to initiate their response”. For the response window, was the 100 ms target stimulus present? Again, where were responses delivered?

Response:

Thank you for pointing out the issue with the unclear expression. We have revised the original manuscript as follows:

L.190 – 194

Participants were directed to press the left button of mouse using their dominant index finger as quickly as possible upon the presentation of the target. The target appeared for 100 ms, then was replaced by a blank grey background. This grey background remained until the participant responded, or for a maximum of 1200 ms, to ensure a timely response.

10) L.192 “target presentation to the execution of the motor response” That would be the motor times. Reaction times do not entirely depend on the speed of motor responses but on the reaction to a perceived variable. In this regard, the authors do not clarify what the initial position of the response hand was (assuming responses were delivered with a hand). Here, reaction times should have been the times from the appearance of the target to the movement initiation. Moreover, since there is a group of elders, some other response metrics should have been considered, e.g., eye movements, and changes in pupil diameter.

Response:

First, we apologize for any misunderstanding caused by our expression.

Reaction time (RT) refers to the interval between the onset of a stimulus and the participant's response, which is a crucial and widely used measure in studies about temporal expectation (Breska & Deouell, 2017; Breska & Ivry, 2018; Chauvin, Gillebert, Rohenkohl, Humphreys, & Nobre, 2016; Droit-Volet, Lorandi, & Coull, 2019; Sanabria, Capizzi, & Correa, 2011; Zanto et al., 2011).

A brief list of descriptions about reaction time from previous studies is as follows:

[1] RT in the prediction tasks refers to the time between onset of the response stimulus and the motor response (Coull, Davranche, Nazarian, & Vidal, 2013).

[2] The RT in both tasks refers to the time between the onset of the target and the motor response (Ren et al., 2019).

[3] The RT was defined as the time duration between the initiation of target and the first observable response in both tasks (Xu, Ren, Misaki, Wu, & Lu, 2021).

Referring to the above literature, we acknowledge that our initial description might have been unclear, leading to misunderstandings. We have made the following modifications to the original manuscript:

L.205 – 206

The RT signifies the duration from the initiation of the target presentation to the motor response.

Additionally, as you suggested, some other response metrics, e.g., eye movements and changes in pupil diameter, are indeed insightful suggestions. We greatly appreciate your profound advice and will consider implementing a different measurement method in our future research to further validate our findings.

· Breska, A., & Deouell, L. Y. (2017). Neural mechanisms of rhythm-based temporal prediction: Delta phase-locking reflects temporal predictability but not rhythmic entrainment. PLoS biology, 15(2), e2001665. 

· Breska, A., & Ivry, R. B. (2018). Double dissociation of single-interval and rhythmic temporal prediction in cerebellar degeneration and Parkinson’s disease. Proceedings of the National Academy of Sciences, 115(48), 12283-12288. 

· Chauvin, J. J., Gillebert, C. R., Rohenkohl, G., Humphreys, G. W., & Nobre, A. C. (2016). Temporal orienting of attention can be preserved in normal aging. Psychology and aging, 31(5), 442. 

· Coull, J. T., Davranche, K., Nazarian, B., & Vidal, F. (2013). Functional anatomy of timing differs for production versus prediction of time intervals. Neuropsychologia, 51(2), 309-319. 

· Droit-Volet, S., Lorandi, F., & Coull, J. T. (2019). Explicit and implicit timing in aging. Acta Psychologica, 193, 180-189. 

· Ren, Y., Xu, Z., Wu, F., Ejima, Y., Yang, J., Takahashi, S., . . . Wu, J. (2019). Does temporal expectation driven by rhythmic cues differ from that driven by symbolic cues across the millisecond and second range? Perception, 48(6), 515-529. 

· Sanabria, D., Capizzi, M., & Correa, Á. (2011). Rhythms that speed you up. Journal of Experimental Psychology: Human Perception and Performance, 37(1), 236. 

· Xu, Z., Ren, Y., Misaki, Y., Wu, Q., & Lu, S. (2021). Effect of Tempo on Temporal Expectation Driven by Rhythms in Dual-Task Performance. Frontiers in Psychology, 12, 755490. 

· Zanto, T. P., Pan, P., Liu, H., Bollinger, J., Nobre, A. C., & Gazzaley, A. (2011). Age-related changes in orienting attention in time. Journal of Neuroscience, 31(35), 12461-12470. 

· 

11) Results: Given the small size of groups, it would be optimal to observe the participants' performance in a box plot and not the overall population result in a bar plot. Line 214 (see Fig 2). It is hard to believe that variance did not increase with tempo conditions. Again, can the authors please present box plots where individuals' performance can be observed?

Response:

Thank you for your valuable suggestion. Following your advice, we have created the box plot shown below:

Additionally, in the second paragraph of our manuscript's original discussion, we focused on exploring this issue. Based on your suggestions, we have further refined our discussion, adding the following content to our original discussion.

L.269 – 272

This observation can be partly interpreted as the shorter interval enabling more accurate and confident anticipation of the target's onset moment. Conversely, a longer interval leads to predictions that are more variable and less precise, accompanied by increased uncertainty, which in turn results in a slower RT [49].

Regarding the small group sizes, as previously mentioned, the number of participants was based on previous related studies and G-power analysis. In response to your insightful suggestions and in alignment with the data analysis methods and graphical formats of related studies, we have updated Fig 2 and Fig 3 as follows. These revisions not only allow for the observation of individual performances but also effectively highlight our key findings: both older and younger participants exhibited temporal expectation effects under moderate tempo conditions, while these effects were observed only in younger participants under fast and slow tempo conditions.

Fig 2. Mean RTs for the younger and older groups in each tempo condition (fast, moderate, slow), collapsed across rhythm condition. The error bars represent the standard error of the mean. Statistical significance is denoted by ***p<0.001 and **p < 0.01.

Fig 3. Mean RT as a function of the tempo condition (fast, moderate, and slow) and rhythm (regular and irregular) for the younger (a) and older (b) groups. The error bars represent the standard error of the mean. Statistical significance is denoted by ***p < 0.001, ** p < 0.01, * p < 0.05.

We hope that these modifications will enhance the quality of the original figures. Thank you once again for your valuable and helpful suggestions. If the current manuscript still does not meet the requirements, we eagerly await your further guidance.

12) Line 223. The effect of each of the conditions within each group seems a consequence of presenting blocks of trials of each condition separately. Again, were conditions delivered in blocks of consecutive trials?

Response:

Thank you for your question regarding the presentation of experimental conditions. Just as we revised in the previous question “These experimental blocks consisted of three tempo categories: 5 fast tempo blocks, 5 moderate tempo blocks, and 5 slow tempo blocks. The presentation order of these experimental blocks was randomly intermixed.” The sequence of the 15 experimental blocks under the three tempo conditions was thoroughly intermingled. The three tempo conditions were conducted block-wise rather than trial-by-trial, aiming to reduce the continuous switching from one tempo to another, which might result in additional difficulties beyond temporal processing itself for the elderly (Correa, Lupiáñez, & Tudela, 2006). Therefore, a considerable number of studies manipulated temporal expectancy between blocks (Á. Correa, Cona, Arbula, Vallesi, & Bisiacchi, 2014; Mónica Trivino, Arnedo, Lupiánez, Chirivella, & Correa, 2011; Monica Trivino, Correa, Arnedo, & Lupianez, 2010). Additionally, the presentation order of these three tempo condition blocks was randomly intermixed. We are of the opinion that this approach likely does not have a significant impact on our results. We hope our response addresses your concerns, but we remain open to further discussions should you have any additional questions or insights.

· Correa, Á., Cona, G., Arbula, S., Vallesi, A., & Bisiacchi, P. (2014). Neural dissociation of automatic and controlled temporal preparation by transcranial magnetic stimulation. Neuropsychologia, 65, 131-136. 

· Correa, A., Lupiáñez, J., & Tudela, P. (2006). The attentional mechanism of temporal orienting: Determinants and attributes. Experimental Brain Research, 169, 58-68. 

· Mento, G., Tarantino, V., Vallesi, A., & Bisiacchi, P. S. (2015). Spatiotemporal neurodynamics underlying internally and externally driven temporal prediction: a high spatial resolution ERP study. Journal of cognitive neuroscience, 27(3), 425-439. 

· Trivino, M., Arnedo, M., Lupiánez, J., Chirivella, J., & Correa, Á. (2011). Rhythms can overcome temporal orienting deficit after right frontal damage. Neuropsychologia, 49(14), 3917-3930. 

· Trivino, M., Correa, A., Arnedo, M., & Lupianez, J. (2010). Temporal orienting deficit after prefrontal damage. Brain, 133(4), 1173-1185. 

Finally, we sincerely thank you for your valuable and insightful suggestions. We have diligently made point-by-point revisions to the original manuscript in accordance with your recommendations. We believe these modifications have enhanced the manuscript, making it more comprehensible and reader-friendly. If the current manuscript still does not meet the standard and needs further revision, we look forward to your patient guidance."

---

## [Decision Letter · Decision Letter 1]

3 Jan 2024

Effect of tempo on the age-related changes in temporal expectation driven by rhythms

PONE-D-23-26702R1

Dear Dr. Xu,

We’re pleased to inform you that your manuscript has been judged scientifically suitable for publication and will be formally accepted for publication once it meets all outstanding technical requirements.

Kind regards,

Enrique Hernandez-Lemus, Ph.D.

Academic Editor

PLOS ONE

Additional Editor Comments (optional):

Reviewers' comments:

Reviewer's Responses to Questions

**Comments to the Author**

1. If the authors have adequately addressed your comments raised in a previous round of review and you feel that this manuscript is now acceptable for publication, you may indicate that here to bypass the “Comments to the Author” section, enter your conflict of interest statement in the “Confidential to Editor” section, and submit your "Accept" recommendation.

Reviewer #1: All comments have been addressed

2. Is the manuscript technically sound, and do the data support the conclusions?

Reviewer #1: Yes

3. Has the statistical analysis been performed appropriately and rigorously? 

Reviewer #1: Yes

4. Have the authors made all data underlying the findings in their manuscript fully available?

Reviewer #1: Yes

5. Is the manuscript presented in an intelligible fashion and written in standard English?

Reviewer #1: Yes

6. Review Comments to the Author

Reviewer #1: There are some interesting bimodal distributions within groups. However, they are not easy to interpret, so I suggest discussing such a phenomenon regardless of being satisfied with the authors' responses.

Similarly, I won't restrain the ms because of definitions. However, regarding RTS, pressing a mouse button is not the most optimal way of measuring the “motor response.” For instance, pressing a button requires two antagonistic movements, i.e., here, releasing the finger and then lowering it down to press the button. The reaction time is related to releasing the finger, since the stimulus triggered that response; this is important because a considerable variation can be observed among participants between young and old people groups. That said, I would suggest a slight change in lines L.205 – 206: “The RT signifies the duration from the initiation of the target presentation to the PRESSING OF THE MOUSE'S BUTTON”.

7. PLOS authors have the option to publish the peer review history of their article (what does this mean?). If published, this will include your full peer review and any attached files.

Reviewer #1: No

---

## [Editor Report · Acceptance letter]

30 Jan 2024

PONE-D-23-26702R1 

PLOS ONE

Dear Dr. Xu, 

I'm pleased to inform you that your manuscript has been deemed suitable for publication in PLOS ONE. Congratulations! Your manuscript is now being handed over to our production team.

Kind regards, 

on behalf of

Prof. Enrique Hernandez-Lemus 

Academic Editor

PLOS ONE